# Unleashing Fungicidal Forces: Exploring the Synergistic Power of Amphotericin B-Loaded Nanoparticles and Monoclonal Antibodies

**DOI:** 10.3390/jof10050344

**Published:** 2024-05-10

**Authors:** Carla Soares de Souza, Victor Ropke da Cruz Lopes, Gabriel Barcellos, Francisco Alexandrino-Junior, Patrícia Cristina da Costa Neves, Beatriz Ferreira de Carvalho Patricio, Helvécio Vinícius Antunes Rocha, Ana Paula Dinis Ano Bom, Alexandre Bezerra Conde Figueiredo

**Affiliations:** 1Laboratório de Tecnologia Imunológica (LATIM), Bio-Manguinhos, Fundação Oswaldo Cruz, Rio de Janeiro 21040-900, Brazil; carla.soares@bio.fiocruz.br (C.S.d.S.); pcristina@bio.fiocruz.br (P.C.d.C.N.);; 2Programa de Pós-Graduação em Pesquisa Translacional em Fármacos e Medicamentos, Farmanguinhos, Fundação Oswaldro Cruz (Fiocruz), Rio de Janeiro 21040-900, Brazilhelvecio.rocha@fiocruz.br (H.V.A.R.); 3Laboratório de Micro e Nanotecnologia, Farmanguinhos, Fundação Oswaldro Cruz (Fiocruz), Rio de Janeiro 21040-361, Brazil; 4Laboratório de Inovação Farmacêutica e Tecnológica, Instituto Biomédico, Universidade Federal do Estado do Rio de Janeiro (Unirio), Rio de Janeiro 22290-250, Brazil; beatriz.patricio@unirio.br

**Keywords:** nanoparticles, monoclonal antibody, amphotericin B, fungicidal

## Abstract

Fungal infections cause 1.7 million deaths annually, which can be attributed not only to fungus-specific factors, such as antifungal resistance and biofilm formation, but also to drug-related challenges. In this study, the potential of Amphotericin (AmB) loaded polymeric nanoparticles (AmB-NPs) combined with murine monoclonal antibodies (mAbs) (i.e., CC5 and DD11) was investigated as a strategy to overcome these challenges. To achieve this goal, AmB-NPs were prepared by nanoprecipitation using different polymers (polycaprolactone (PCL) and poly(D,L-lactide) (PLA)), followed by comprehensive characterization of their physicochemical properties and in vitro biological performance. The results revealed that AmB-loaded NPs exhibited no cytotoxicity toward mammalian cells (baby hamster kidney cells—BHK and human monocyte cells—THP-1). Conversely, both AmB-NPs demonstrated a cytotoxic effect against *C. albicans*, *C. neoformans*, and *H. capsulatum* throughout the entire evaluated range (from 10 µg/mL to 0.1 µg/mL), with a significant MIC of up to 0.031 µg/mL. Moreover, the combination of AmB-NPs with mAbs markedly intensified antifungal activity, resulting in a synergistic effect that was two to four times greater than that of AmB-NPs alone. These findings suggest that the combination of AmB-NPs with mAbs could be a promising new treatment for fungal infections that is potentially more effective and less toxic than current antifungal treatments.

## 1. Introduction

Invasive fungal infections (IFIs) are prominent etiological determinants of worldwide morbidity and mortality [1]. According to the Global Action Fund for Fungal Infections (GAFFI), the annual global incidence of individuals with fungal infections exceeds 300 million cases [2]. Forecasts within this group portend a mortality toll surpassing 1.6 million deaths, which parallels the concerning impact of tuberculosis and concurrently manifests a threefold malaria-associated fatality [3]. The increase in the incidence of fungal infections can be attributed to a combination of determinants, including the pervasive use of immunosuppressive agents, antineoplastic compounds, and broad-spectrum antibiotics [4]. Moreover, individuals harboring HIV exhibit a notable propensity to develop fungal infections, which is exemplified by the emergence of severe opportunistic mycoses caused by pathogens such as *Candida albicans*, *Aspergillus fumigatus*, and *Cryptococcus neoformans*. In tandem, the global panorama of AIDS cases exhibits an upward trajectory, leading to a mortality quotient that spans within the spectrum of 50% [5].

The management of these IFIs with amphotericin B (AmB) has achieved favorable outcomes. This polyene macrolide antifungal agent has been used clinically since the latter half of the 1950s, and it remains the primary pharmaceutical intervention for fungal infections [6]. Nevertheless, AmB exhibits significant challenges stemming from its low aqueous solubility, erratic oral absorption, and important cytotoxic effects, including acute renal failure [7]. As a result, its clinical use is limited to intravenous administration, where the currently available treatment imposes a cost-prohibitive burden [8] and has an elevated cost-effectiveness ratio [9].

In light of this scenario, research has been conducted with the primary impetus of developing alternative formulations for effective oral administration of AmB [10]. This administration route confers several advantages, including greater patient acceptance and treatment adherence, in addition to being safer and less expensive, enabling access to the medicine for a more significant portion of the population [11]. Among the many approaches currently available, the use of polymeric nanoparticles (NPs) has many advantages, such as a substantial capacity for drug solubilization and encapsulation, protection against chemical and enzymatic degradation in the gastrointestinal tract, biocompatibility, high stability, controlled release, functionalization, and potential for implementing staggered production on an industrial scale [12].

Although AmB has demonstrated remarkable efficacy for several decades, fungal resistance has been reported, and serious concerns have been raised regarding its potential negative impact on the public health system [13]. No class of antifungals has been introduced on the market since 2006, when the European Medicines Agency (EMA) and the Food and Drug Administration (FDA) approved anidulafungin [14,15]. Monoclonal antibodies are powerful tools currently used for diagnosis but have not yet been developed for the treatment of invasive fungal infections. Research has demonstrated that monoclonal antibodies can sensitize fungal cells, thereby enhancing the fungicidal efficacy of drugs [16,17]. This phenomenon has the potential to be a viable therapeutic strategy to overcome fungal resistance. Therefore, this study systematically investigated the prospective utility of AmB-loaded polymeric nanoparticles (AmB-NPs) in conjunction with murine monoclonal antibodies (mAbs), i.e., CC5 and DD11, within an in vitro framework. The primary objective is to assess the viability of this approach as a strategic avenue for surmounting these limitations.

## 2. Materials and Methods

### 2.1. Polymeric Nanoparticles: Preparation and Physicochemical Characterization

The polymeric nanoparticles (NPs) were obtained by the nanoprecipitation method (Figure 1), according to the methodology previously described by Marcano et al. [18]. Briefly, a polymeric solution of either PCL or PLA was prepared in acidified acetone, with heat applied (40 °C) to facilitate PCL solubilization (RT 15 Power IkaMag, IKA, Staufen, Germany). In parallel, a 1% *m*/*v* AmB solution was prepared in DMSO, followed by the addition of MetOH (1:2.5 ratio). Subsequently, this organic phase was carefully added to an aqueous solution containing 0.3% *m*/*v* polysorbate 80 via magnetic stirring at room temperature. Afterward, the volatile organic solvents were removed using a rotary evaporator (RV9, IKA, Staufen, Germany) at 40 °C for 30 min. To ensure the removal of the untrapped AmB, centrifugation was carried out in 1 h cycles at 3000 rpm and room temperature (Megafuse 16, Thermo Fisher Scientific, Waltham, MA, USA). To ensure the reliability of the experimental results, all amphotericin B-loaded nanoparticles (AmB-NPs) were freshly prepared prior to any subsequent evaluation. This approach addressed the well-documented instability of AmB in aqueous media, mitigating potential degradation that could compromise the performance of the nanoparticles.

These formulations were characterized in terms of hydrodynamic diameter (z-average), polydispersity index (PDI) by dynamic light scattering (DLS), and surface charge (zeta potential) by measuring electrophoretic mobility using a Zetasizer Nano ZS90 (Malvern, Worcestershire, UK) at 25 °C. A 1:50 dilution (NP: Milli-Q water) was carried out before each measurement, and the aforementioned parameters were calculated from the average of readings performed in triplicate for each of the samples, as well as the standard deviation. The results were obtained using ZetaSizer software 7.11, and the statistical analyses were performed with GraphPad Prism 8.0.1 software.

The amount of AmB within the nanoparticles was measured by UV–Vis spectrophotometry (UV 1800, Shimadzu, Kyoto, Japan) using a previously validated method. Briefly, 500 μL of sample was carefully transferred to a microcentrifuge tube equipped with a 100 kDa Amicon filter, followed by 20 min of centrifugation at 7500× *g*. Subsequently, the unfiltered material was appropriately diluted with acetonitrile:DMSO (6:4), and the absorbance at λ = 411 nm was recorded.

The morphology of the obtained nanoparticles was analyzed on a JEM-1230 transmission electron microscope (JEOL Ltd., Tokyo, Japan). Ten microliters of each sample was dripped onto Formvar-coated grids for 1 min and subsequently incubated with 2% *m*/*v* uranyl acetate. The excess contrast solution was removed from the grid with filter paper, and the samples were kept at room temperature until analysis.

### 2.2. Strains and Growth Conditions

The fungal strains used in this study were *C. neoformans* (H99), *C. albicans* (ATCC 90028) and *Histoplasma capsulatum* (ATCC 26032). The microorganisms were maintained at 30 °C in Sabouraud broth (*C. neofomans*) or liquid brain heart infusion (BHI, *C. albicans*), and *Histoplasma capsulatum* was grown in Ham’s F-12 medium supplemented with glucose (16 g/L), glutamic acid (1 g/L), HEPES (6 g/L) and cysteine (8.4 mg/mL) at 37 °C.

### 2.3. Cultivation of Human and Mammalian Cells

Two cell lines, namely, BHKs (ATCC: CCL-10) and THP-1 cells (ATCC: TIB-202), were used in this study. These cells were cultivated in RPMI containing 10% fetal bovine serum (FBS) at 37 °C in an atmosphere of 5% CO_2_. BHK cells were trypsinized with trypsin-EDTA solution for 5 min at 37 °C. Cell counts were carried out in a Neubauer chamber.

### 2.4. Cell Viability Assay

NP toxicity was evaluated by assessing the capacity of viable cells to metabolize methylthiazolyldiphenyl-tetrazolium bromide (MTT) [19]. Briefly, BHK and THP-1 cells (10^6^/100 μL of RPMI 1640) were incubated for 24 h at 37 °C with different concentrations of AmB-loaded NPs (10–0.3 µg/mL). Comparatively, *C. neoformans*, *C. albicans*, and *H. capsulatum* (10^6^/100 μL of RPMI 1640) were cultivated under the same conditions, i.e., 24 h at 37 °C, and exposed to NPs with or without AmB at concentrations ranging from 2–0.0015 µg/mL. For the death control, 10% DMSO was used for mammalian cells, and 2 µg/mL AmB was used for fungal cells. Subsequently, the cells were incubated with 25 μL of MTT (5 mg/mL in PBS) for 4 h at 37 °C in the dark. Afterward, the MTT solution was carefully removed, and the cells were exposed to 100 μL of DMSO protected from light to ensure the complete solubilization of the formazan crystals. The outcomes were quantified through spectrophotometric analysis at λ = 540 nm.

### 2.5. Antifungal Activity

Antimicrobial tests were carried out using the EUCAST protocol adapted for yeast cells [20]. *C. neoformans*, *C. albicans*, and *H. capsulatum* were inoculated in RPMI 1640 supplemented with 2% glucose and adjusted to pH 7.0. The *C. neoformans* cells were inoculated at 10^7^ cells/mL, and the other fungal cells were inoculated at 10^6^ cells/mL in 96-well plates at a final volume of 200 μL. Free AmB was used as a control at both inhibitory and subinhibitory concentrations (2 and 0.1 μg/mL, respectively). After 48 h of incubation at 37 °C with continuous agitation, the cells were thoroughly homogenized using a pipette and subsequently subjected to spectrophotometric analyses at λ = 592 nm.

### 2.6. Synergistic Effect of Anti-Chitooligomer Antibodies with NPs

The monoclonal antibodies were produced following the same conditions described previously [16]. To assess the effect of associating the anti-chitooligomer antibodies with NPs, the cells were inoculated at the same concentration as those used for antifungal activity tests. Therefore, the NP minimal effective concentration was established before investigating a spectrum of murine antibody concentrations (25–0.3 µg/mL) against *C. neoformans* and *C. albicans* [16]. The synergistic potential of the AmB-loaded NPs, both alone and in combination with the mAbs, was evaluated by calculating the fractional inhibitory index (FII). The categorization of synergistic effects was determined as follows: FII < 1 denoted a synergistic effect, while FII = 1 indicated an additive effect [21].

### 2.7. ELISA Analysis of Glucuronoxylomannan (GXM) in C. neoformans

ELISA plates (96-well) were coated with culture supernatant obtained from MIC treatment (48 h) and incubated overnight at 4 °C. Subsequently, the plate was incubated with PBS containing 1% BSA for 1 h at 37 °C, followed by washing with PBS containing 0.05% Tween. The 2D10 monoclonal antibody was added at a concentration of 1 µg/mL and incubated for 2 h at 37 °C. The plate was washed three times with PBS-Tween, after which anti-murine IgG conjugated to peroxidase was added and incubated for 2 h at 37 °C. Subsequently, the plates were washed as described previously and incubated with tetramethylbenzidine (TMB) for 30 min at 37 °C. The reaction was quenched with 1 M HCl, and spectrophotometric readings were obtained at λ = 450 nm. The reactions were considered positive if the absorbance values were 3 times greater than the cutoff without the addition of primary antibodies.

### 2.8. Statistical Analysis

All the statistical analyses were performed using GraphPad Prism software, version 8.0.1. Two-way ANOVA with Bonferroni post hoc correction was used for individual comparisons between groups, and a 95% confidence interval was used for all experiments.

## 3. Results

### 3.1. Physicochemical Characterization of the NPs

The nanoprecipitation method produced monodisperse NPs as depicted in Figure 2, evidenced by the low PDI values (<0.2), a zeta potential close to −20 mV (Figure 3) and exhibiting a relatively spherical shape (Figure 4). Statistical analysis indicated that although the particle size did not change significantly with AmB encapsulation (*p* > 0.05), the polymer used to manufacture the NPs significantly affected their size (*p* < 0.05), regardless of the presence of AmB. Moreover, no significant difference in AmB encapsulation was detected between the NPs, with PLA-AmB and PCL-AmB having comparable AmB loadings (*p* > 0.05), i.e., 134 ± 18 mg/mL and 139 ± 12 mg/mL, respectively.

### 3.2. Cytotoxicity of NPs

Polymeric nanoparticles loaded with AmB did not exhibit cytotoxicity at any concentration evaluated (10–0.3 μg/mL) when tested against mammalian cells (i.e., BHK and THP-1 cells). A similar outcome was found in fungal cells treated with NPs without AmB (Figure 5 and Figure 6). Nevertheless, when AmB-loaded NPs were tested against fungi, toxicity was clearly observed in all fungal species tested (Figure 7), with PCL-AmB and PLA-AmB exhibiting cytotoxic effects across a concentration range of 10 to 0.025 µg/mL (*p* < 0.05) and 10 to 0.1 µg/mL (*p* < 0.05), respectively.

### 3.3. Antifungal Activity

#### 3.3.1. Antifungal Activity of AmB-NPs

The potential antifungal activity of the NPs against *C. neoformans*, *C. albicans*, and *H. capsulatum* was assessed using standardized methods recommended by the European Committee on Antimicrobial Susceptibility Testing (EUCAST) [20]. PLA and PCL nanoparticles had no discernible impact on fungal growth when tested independently. However, once AmB was encapsulated, a prominent antifungal effect became evident for both the PLA and PCL-AmB formulations, and for PLA-AmB against *C. neoformans*, this effect ranged from 2 to 0.5 μg/mL (MIC 90%). Similar results were observed when these formulations were tested against *C. albicans* and *H. capsulatum*. However, it is important to note that an MIC of 90% was observed within the concentration range of 2 to 0.125 μg/mL for both NP formulations against *C. albicans* and 2 to 0.031 μg/mL against *H. capsulatum*, while no discernible effect was observed at lower concentrations (Figure 8).

#### 3.3.2. Antifungal Activity of AmB-Loaded NPs in Combination with mAbs

To assess the synergistic effect of AmB-loaded NPs with mAbs, freshly prepared nanoparticles containing noninhibitory concentrations of PLA-AmB and PCL-AmB formulations (i.e., 0.06 μg/mL for *C. neoformans* and 0.03 μg/mL for *C. albicans*) were supplemented with mAbs. As previously mentioned, no discernible inhibition of fungal growth was observed when AmB-loaded NPs were tested independently, i.e., without the addition of mAbs (Figure 6).

However, when mAbs were added to the AmB-loaded NP formulations, notable inhibition of fungal growth was observed for both antibodies at varying concentrations (Figure 9, Figure 10, Figure 11 and Figure 12). Furthermore, the interaction between the mAbs and AmB-loaded NPs was assessed by calculating the fractional inhibitory index (FII) [21], revealing a synergistic interaction between these components (*C. neoformans*—PCL or PLA + CC5 (25 µg/mL) = FII < 0.5; *C. neoformans*—PCL + DD11 (12.5 µg/mL) = FII < 0.5; *C. neoformans*—PLA + DD11 (25 µg/mL) = FII < 0.5; *C. albicans*—PCL or PLA + CC5 or DD11 (0.3 µg/mL) = FII < 0.5).

#### 3.3.3. Effect of mAbs in Synergism with NP-AmB on GXM Release

A synergistic impact on fungal growth was observed when mAbs were combined with NP-AmB. The influence of this antifungal approach on the release of GXM, an exopolysaccharide involved in numerous immunoregulatory mechanisms, was further investigated. As depicted in Figure 9, Figure 10, Figure 11 and Figure 12, after 48 h of fungal cell culture, the concentration of GXM was directly positively correlated with the mAb concentration in the culture medium (Figure 13).

## 4. Discussion

The increase in antifungal resistance has led to the need for synergistic combinations of multiple drugs to achieve promising outcomes. A potential therapeutic strategy to overcome this alarming scenario would be to combine antifungal agents with monoclonal antibodies [22]. Therefore, the primary objective of this study was to assess the efficacy of combining polymeric nanoparticles (PLA-AmB and PCL-AmB) with murine monoclonal antibodies as a strategy to manage fungal infections caused by PLA.

MTT assays of AmB-loaded NPs revealed that these nanoparticles did not exhibit any discernible toxicity toward mammalian cells (i.e., BHK and THP-1 cells) at any concentration evaluated. Therefore, these findings highlight the biocompatibility of these formulations and the possibility of using them in in vivo treatments. In addition, after assessing the toxicity of nanoparticles without AmB to verify whether these formulations could independently affect fungal growth, no cytotoxic effects were detected for any of the tested fungi, namely, *C. neoformans*, *C. albicans*, or *H. capsulatum*. These findings may indicate that nanoparticles do not have any inherent toxic effects on fungal cells, leading to the conclusion that any toxicity should be attributed to the encapsulated AmB. Furthermore, these findings corroborate the findings of studies of these NPs in mammalian cells, in which nanoparticles without AmB or loaded with AmB did not exhibit toxic effects, and against fungal cells, in which toxicity was observed only when the NPs were loaded [23,24].

Subsequent to assessing AmB-loaded NP cytotoxicity across diverse cell types and fungal species, these nanoparticles were evaluated for their potential to inhibit fungal growth by performing a minimum inhibitory concentration (MIC) test. PCL-AmB and PLA-AmB formulations exhibited a discernible effect on fungal growth in vitro. For *C. neoformans*, both nanoparticles presented MICs lower than 50%, extending to a concentration of 0.25 µg/mL. Notably, while a propensity for reduced growth was evident at the 0.125 µg/mL concentration, this trend did not reach statistical significance (*p* value > 0.05). Notably, AmB-loaded NPs had a more significant effect than nonencapsulated AmB. These data are consistent with the literature that underscores the therapeutic efficacy of the encapsulated formulation [18].

Regarding the MIC for *C. albicans*, a more pronounced effect of AmB-loaded NPs was detected, with MICs ranging from 90% to 50% observed at concentrations up to 0.062 µg/mL. For *H. capsulatum*, the MIC remained consistent at 90% across all concentrations tested, up to 0.031 µg/mL, highlighting the potential to enhance the efficacy of the drug through its formulation at the nanoscale.

Numerous investigations have been conducted to assess the synergistic utilization of drugs, leading to a consequential reduction in individual drug concentrations. This strategy promises to mitigate potential side effects and diminish the likelihood of resistance emergence [25]. In previous studies conducted by our team, coadministration of an anti-chitooligomer monoclonal antibody (mAb) at concentrations of 25, 12.5, and 6.2 µg/mL (CC5 and DD11) in combination with AmB at 0.1 µg/mL exhibited significant antifungal activity both in vitro and in vivo [16]. This finding prompted us to investigate the inhibitory potential of AmB-loaded NPs at different concentrations (0.03 µg/mL for *C. albicans*/0.06 µg/mL for *C. neoformans*). Building upon our previous research, a synergistic effect was obtained by combining these nanoparticles with the aforementioned mAb, particularly at 25 μg/mL initial concentrations.

This combined approach demonstrated high inhibition capacity, surpassing the individual effect of AmB-loaded NPs. Moreover, this enhanced inhibition against *C. albicans* was sustained up to 0.35 µg/mL at 24 and 48 h (CCC and DD11). This outcome holds significant promise, especially when contrasted with the fungal activity of the anti-chitin monoclonal antibody combined with AmB, in which greater efficacy was achieved using an even lower concentration of AmB than in our previous study, once again demonstrating the effectiveness of AmB-loaded NPs [16].

It is worth highlighting that the synergistic effect of mAbs with AmB-loaded NPs was established across both tested fungal species. Notably, the inhibitory effect on *C. albicans* was more pronounced than that on *C. neoformans* after a 48 h incubation period. This outcome may be attributed to the inherent structural differences between these species, a phenomenon that is well documented in the literature. Specifically, the fungal species *C. neoformans* is characterized by polysaccharide capsules composed of glucuronoxylomannan (GXM) and galactoxylomannan (GalXM) [26,27,28], which are distinct from those of *C. albicans*.

Remarkably, the mAb combined with AmB-loaded NPs augmented the antifungal effect on *C. neoformans*, surpassing the individual effect of AmB-loaded NPs. It is postulated that the mAb may directly interact with the cell wall, potentially influencing its morphology and altering drug action [16]. These findings are consistent with the results of the GXM detection assay in which cultured supernatants were treated with AmB-loaded NPs in isolation and combined with the mAb. The mAb seems to destabilize the cell wall, facilitating the release of crucial structural constituents, such as GXM. In fact, this pronounced GXM release was primarily mediated by the mAb, as AmB-loaded NPs exhibited considerably less of an impact than the synergistic combination (*p* value < 0.05). PLA-AmB seems to destabilize the cell wall more than PCL-AmB; however, it does not have fungicidal effects when synergistically combined, as demonstrated by the MIC assay. This indicates that the action of the mAb increases fungal membrane permeability, thereby enhancing the effectiveness of the AmB-loaded NPs.

In summary, mAbs have shown promising potential as adjuvant agents for the treatment of fungal infections, especially when they are used with polymeric nanoformulations. Additionally, AmB-NPs have shown notable efficacy in augmenting antifungal activity, enhancing overall drug efficacy. This prospect holds the potential to reduce drug dosages and associated adverse effects. Using mAbs to target chitooligomers, a biopharmaceutical component for fungal disease treatment, with lower doses of nanoformulated AmB could reduce healthcare costs related to therapy and hospitalization.

## 5. Future Perspectives

This study aimed to establish a proof of concept by demonstrating the synergistic effect of AmB-loaded NPs in combination with mAbs against fungal cells. The observed data support the potential of these formulations for antifungal therapy. Nevertheless, further development studies are required to translate this concept into a clinically applicable formulation. Freeze-drying, for example, appears to be a promising approach to mitigating the aqueous instability of AmB, potentially improving its long-term storage and transportability. Additionally, in vivo studies, e.g., pharmacokinetic–pharmacodynamic (PK/PD) modeling, can provide valuable insights into the relationship between drug exposure and antifungal activity. Implementing these models will facilitate the optimization of dosing strategies and treatment schedules for optimal therapeutic efficacy.

## 6. Patents

**Transparency declaration**: Part of the data presented here is the subject of pending patent applications (Reference numbers: BR 10 2020 002165 6, WO2021/151180 A2, PCT/BR2021/050007).

## Figures and Tables

**Figure 1 jof-10-00344-f001:**
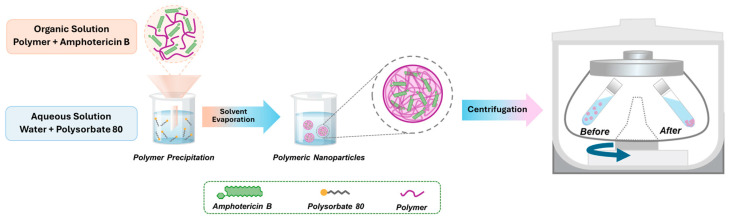
Schematic representation of the nanoprecipitation method used to prepare nanoparticles.

**Figure 2 jof-10-00344-f002:**
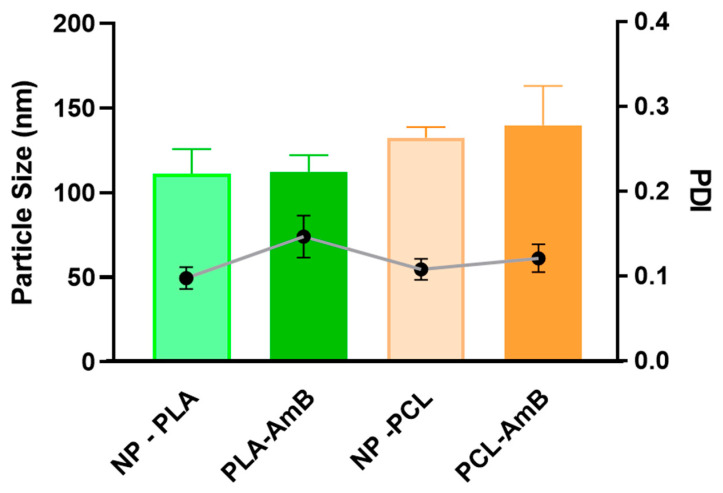
Impact of amphotericin B encapsulation and polymeric matrix on polydispersity and particle size. Note: Amphotericin B (AmB) encapsulation did not significantly alter the particle size or polydispersity index (PDI) of polymeric nanoparticles (NPs) prepared with either polycaprolactone (PCL) or poly(D,L-lactide) (PLA) (*p* > 0.05). Nevertheless, the type of polymer matrix used for NP manufacturing significantly influenced their size (*p* < 0.05). Particle size is represented by the bars, while PDI is denoted by the dots in the graph.

**Figure 3 jof-10-00344-f003:**
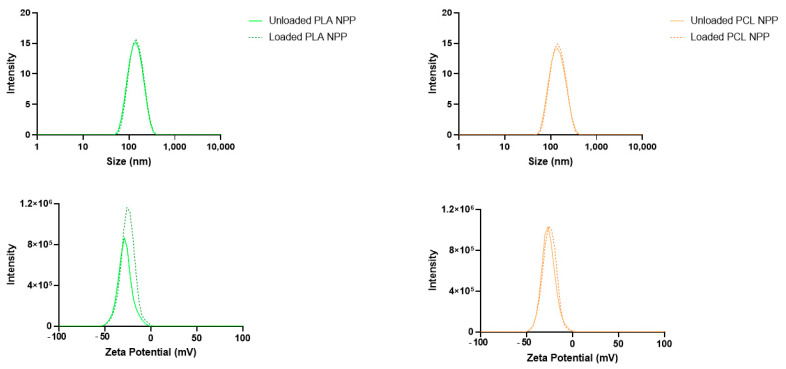
The PCL and PLA NP suspensions exhibited homogeneous particle sizes and zeta potential distributions, regardless of whether they were loaded with AmB.

**Figure 4 jof-10-00344-f004:**
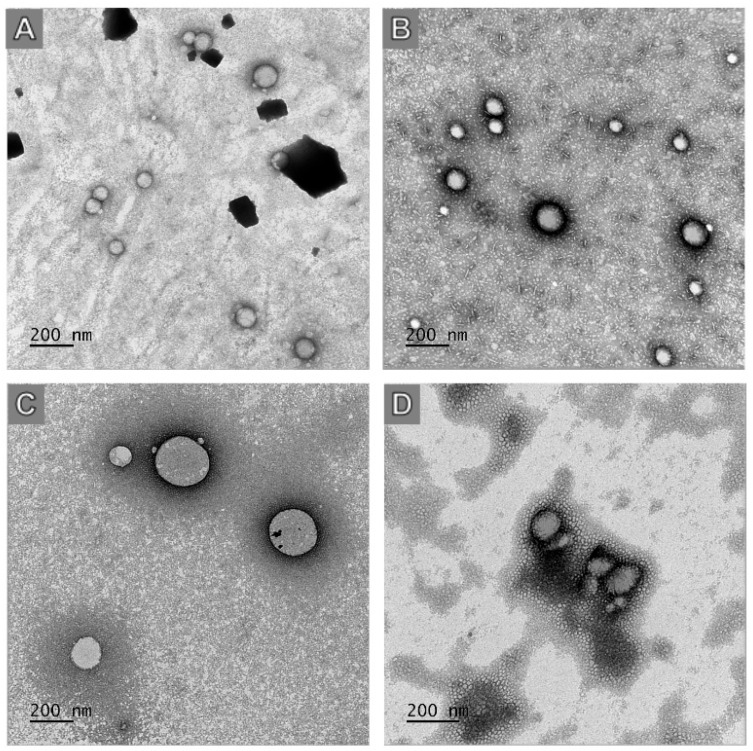
Photomicrographs obtained by transmission electron microscopy of (**A**) NP-PLA, (**B**) PLA-AmB, (**C**) NP-PCL and (**D**) PCL-AmB with 200 nm resolution.

**Figure 5 jof-10-00344-f005:**
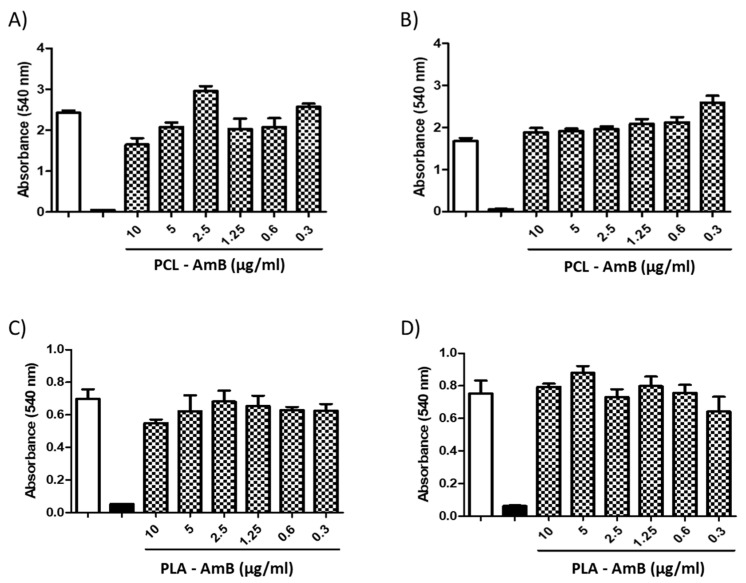
Cytotoxicity of AmB-loaded NPs to BHK and THP-1 cells. PCL-AmB and PLA-AmB were tested against BHK cells (**A**,**B**) and against THP-1 cells (**C**,**D**), and no measurable cytotoxic effect was observed at concentrations ranging from 10 to 0.3 µg/mL. The white and black columns correspond to the negative control (untreated) and positive control (100% death—treated with 10% DMSO), respectively.

**Figure 6 jof-10-00344-f006:**
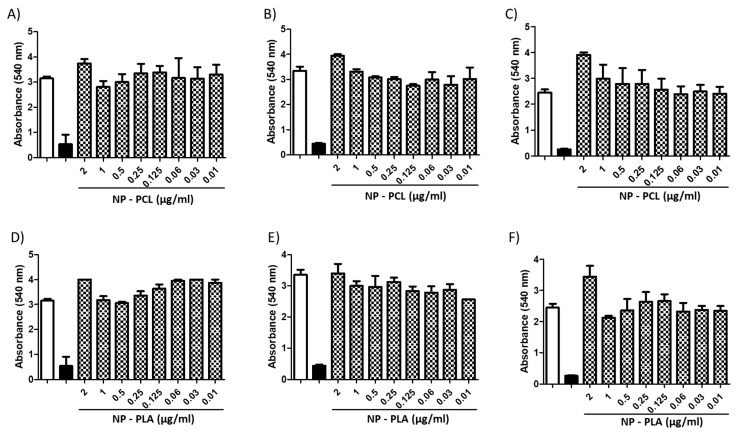
Cytotoxicity assay with NPs without AmB. No toxic effects were detected for either NP-PCL or NP-PLA against *C. neoformans* (**A**,**D**), *C. albicans* (**B**,**E**) or *H. capsulatum* (**C**,**F**) in the concentration range of 2–0.01 μg/mL. The white and black columns correspond to the negative control (untreated) and positive control (100% death—2 µg/mL AmB), respectively.

**Figure 7 jof-10-00344-f007:**
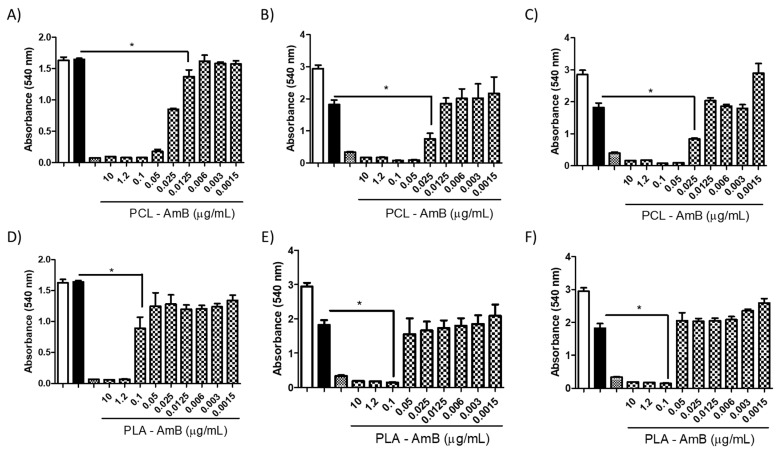
Cytotoxicity assay with AmB-loaded NPs against *C. neoformans* (**A**,**D**), *C. albicans* (**B**,**E**), and *H. capsulatum* (**C**,**F**), demonstrating the toxicity of both NP PCL-AmB (**A**–**C**) and PLA-AmB (**D**–**F**) at concentrations ranging from 10 to 0.0015 µg/mL (* *p* < 0.05). The white, black, and dotted white columns represent the following controls: untreated; AmB 0.1 μg/Ml; and AmB 2 µg/mL (100% death).

**Figure 8 jof-10-00344-f008:**
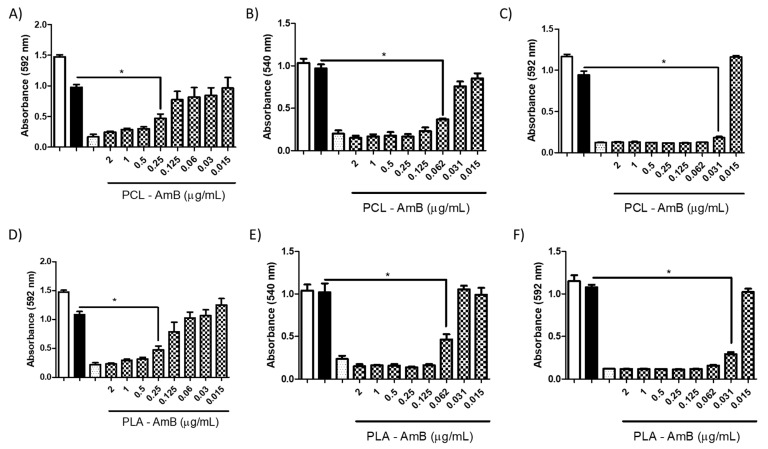
Minimum inhibitory concentration (MIC) of PCL-AmB and PLA-AmB against fungal species. An inhibitory effect was observed at concentrations of 0.25 μg/mL (* *p* < 0.05) for *C. neofomans* (**A**,**D**), 0.062 μg/mL (*p* < 0.05) for *C. albicans* (**B**,**E**), and 0.031 μg/mL (* *p* < 0001) for *H. capsulatum* (**C**,**F**). The control groups included the AmB 2 μg/mL (dotted white), AmB 0.1 μg/mL (black) and untreated (white) groups.

**Figure 9 jof-10-00344-f009:**
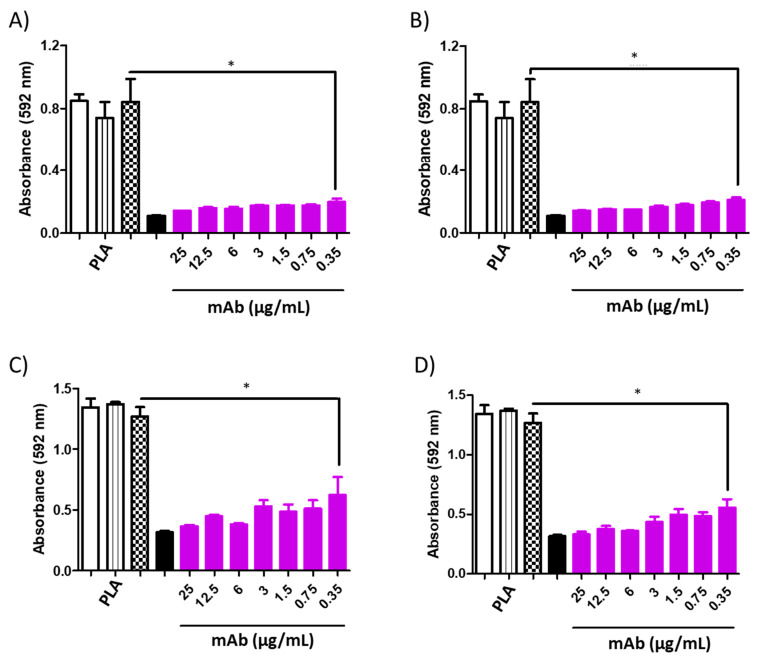
Minimum inhibitory concentration (MIC) of PLA-AmB (0.3 μg/mL) in combination with the murine monoclonal antibodies (mAbs) DD11 ((**A**)—24 h and (**C**)—48 h) and CC5 ((**B**)—24 h and (**D**)—48 h) against *C. albicans*. An inhibitory effect was observed throughout the evaluated range, i.e., from 25 to 0.35 μg/mL (* *p* < 0.05). Controls: untreated (white), 0.03 μg/mL PLA-AmB (black stripes), 0.1 μg/mL free AmB (dotted white), and 2 μg/mL AmB (black).

**Figure 10 jof-10-00344-f010:**
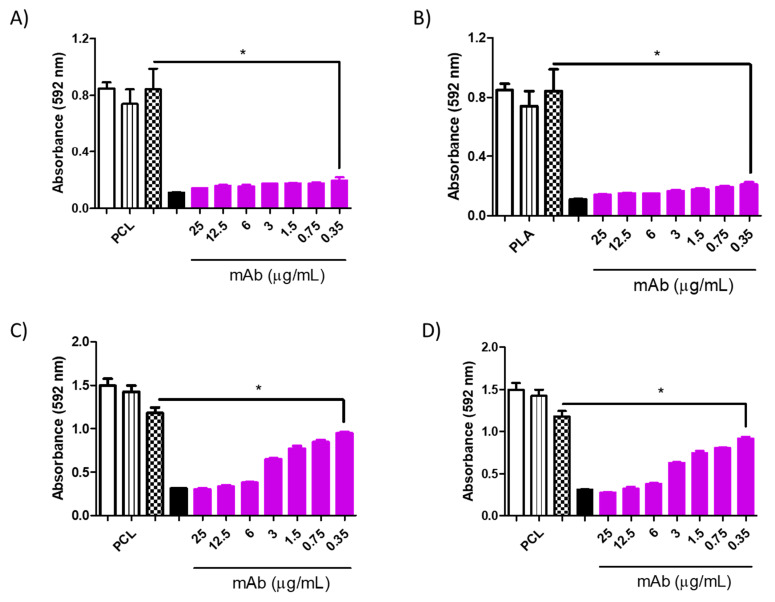
Minimum inhibitory concentration (MIC) of PCL-AmB (0.03 μg/mL) in combination with murine monoclonal antibodies (mAbs DD11 ((**A**)—24 h and (**C**)—48 h) and CC5 ((**B**)—24 h and (**D**)—48 h) against *C. albicans*. The inhibitory effect was observed throughout the evaluated range up to 0.35 μg/mL (* *p* < 0.05). Controls: untreated (white), PCL-AmB 0.03 μg/mL (black stripes), free AmB 0.1 μg/mL (dotted white), and AmB 2 μg/mL (black).

**Figure 11 jof-10-00344-f011:**
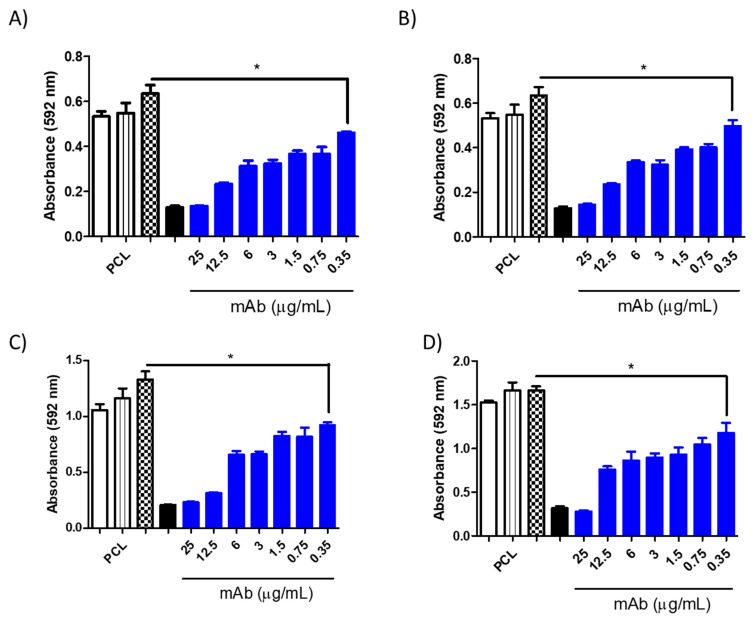
Minimum inhibitory concentration (MIC) of PCL-AmB (0.06 μg/mL) in combination with the murine monoclonal antibodies (mAbs) DD11 ((**A**)—24 h and (**C**)—48 h) and CC5 ((**B**)—24 h and (**D**)—48 h) against *C. neoformans*. The inhibitory effect after 48 h reached 0.35 μg/mL for DD11 and CC5 (* *p* < 0.05). Controls: untreated (white), PCL-AmB 0.03 μg/mL (black stripes), free AmB 0.1 μg/mL (chess), and free AmB 2 μg/mL (black).

**Figure 12 jof-10-00344-f012:**
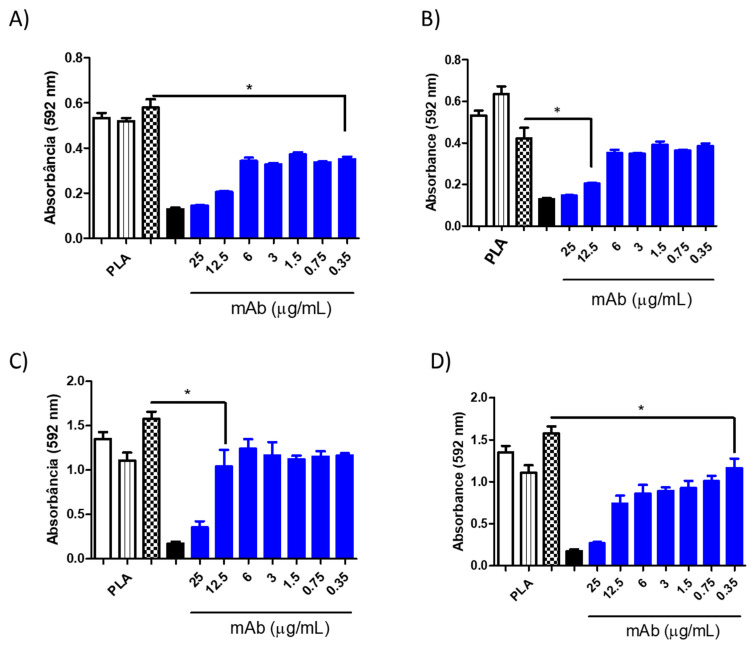
Minimum inhibitory concentration (MIC) of PLA-AmB (0.06 μg/mL) in combination with the murine monoclonal antibodies (mAbs) DD11 ((**A**)—24 h and (**C**)—48 h) and CC5 ((**B**)—24 h and (**D**)—48 h) against *C. neoformans*. The inhibitory effect after 48 h reached 0.35 μg/mL for CC5 (* *p* < 0.05 and 12.5 μg/mL for DD11 (* *p* < 0.05). Controls: untreated (white), PCL-AmB 0.03 μg/mL (black stripes), free AmB 0.1 μg/mL (chess), and free AmB 2 μg/mL (black).

**Figure 13 jof-10-00344-f013:**
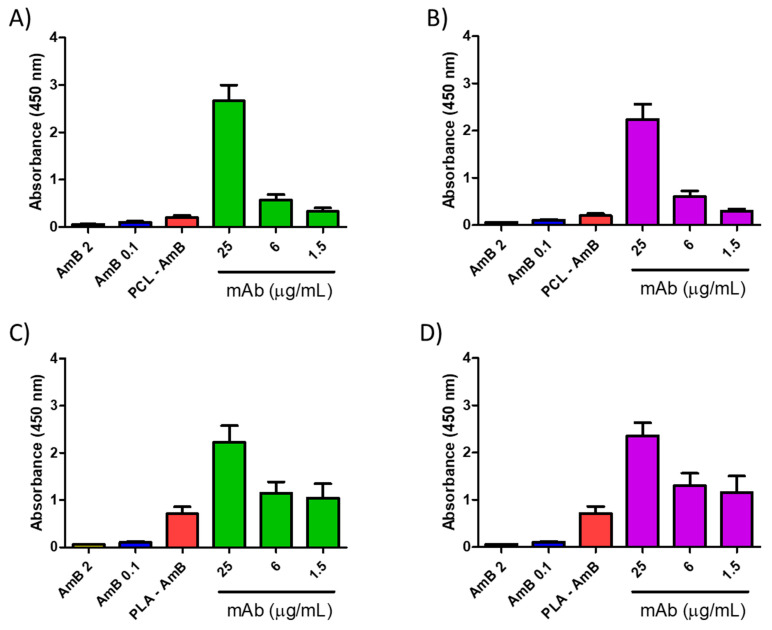
Quantification of glucuronoxylomannan (GXM) release in the supernatant of *C. neoformans* culture by indirect ELISA after 48 h of exposure to polymeric nanoparticles (0.06 μg/mL) prepared with either polycaprolactone (PCL) or poly(D,L-lactide) (PLA) combined with increased concentrations of the murine monoclonal antibodies CC5 ((**A**,**C**)—green) or DD11 ((**B**,**D**)—purple). Controls: Free AmB at 2 μg/mL (black), free AmB at 0.1 μg/mL (blue) and AmB-loaded NPs at 0.06 μg/mL (red).

## Data Availability

Data are contained within the article.

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
