# Peer review of "Unleashing Fungicidal Forces: Exploring the Synergistic Power of Amphotericin B-Loaded Nanoparticles and Monoclonal Antibodies"

_jof, 2024, doi:10.3390/jof10050344_

Round 1

Reviewer 1 Report

This study was carried out well, and adds to the body of knowledge in the field. 

English editing of spelling and grammar is however required before publication.

I suggest a thorough check of English spelling and grammar. Many errors were noted eg. "supernatant" spelled as "supernadant" (line 150)

Author Response

Dear Reviewer,

Thank you about your comments. 

I will check of English spelling and grammar.

Regards,

Alexandre Figueiredo

Reviewer 2 Report

This study is well done; however, the manuscript requires correction.

Section "Materials and Methods".

The authors were careless with this section; there are many unclear points here that need to be eliminated for readers to better understand the manuscript. There are inconsistencies between this section and the Results section; in the “Results” section there is data, the method of obtaining which in “Materials and Methods” is poorly described or not described at all.

This study is well done; however, the manuscript requires correction.

Section "Materials and Methods".

The authors were careless with this section; there are many unclear points here that need to be eliminated for readers to better understand the manuscript. There are inconsistencies between this section and the Results section; in the “Results” section there is data, the method of obtaining which in “Materials and Methods” is poorly described or not described at all.

Subsection “2.4. Cell viability assay", lines 119-130. It indicates how many BHK and THP-1 cells (106/ 100 μL) were used, but nothing about the number of fungal cells. It is necessary to indicate the number of C. neoformans, C. albicans, and H. capsulatum cells used in these experiments. In section “3. Results” in the captions to Figures 2 and 3 (and so on), it is indicated that dead cells were used as one of the controls. How were dead cells obtained?

Subsection “2.5. Antifungal activity", lines 131-139.

References [14] (line 133) and [15] (line 139) must be in square brackets; now references are in the form of a number in the superscript. What doses of amphotericin B-loaded nanoparticles were used here?

Subsection “2.6. Synergism effect of antichitooligomer antibodies with NP", lines 140-148.

Reference [17] (line 148) must be in square brackets.

What is the origin of monoclonal antibodies? What are the characteristics of monoclonal antibodies? Why were mouse antibodies used? The reviewer believes that these questions should be answered in this subsection.

Section 3. Results.

Why did the authors measure the cytotoxicity of nanoparticles without amphotericin B against fungi only? Why wasn't this done for mammalian cells?

Author Response

Dear Reviewer,

The authors would like to express their gratitude to the reviewers for their time and effort in offering thoughtful and insightful feedback on the manuscript. The comments and suggestions provided have notably enhanced the quality of the manuscript. 

 - Subsection “2.4. Cell viability assay", lines 119-130

Authors Response: Thank you for pointing this out. The same concentration of cells was used for both mammalian cells and fungal cells. Regarding the death control used in the experiments, we used 10% for mammalian cells, while for fungal cells AmB was used. 

- Subsection “2.5. Antifungal activity", lines 131-139. What doses of amphotericin B-loaded nanoparticles were used here? 

Authors Response: AmB-NP doses vary from 2 to 0.015 ug/ml as shown in figure 5 of the manuscript, but ww will put in more detail into the methodology.

Subsection “2.6. Synergism effect of antichitooligomer antibodies with NP", lines 140-148. 

Reference [17] (line 148) must be in square brackets. 

What is the origin of monoclonal antibodies?  What are the characteristics of monoclonal antibodies? Why were mouse antibodies used? The reviewer believes that these questions should be answered in this subsection. 

Authors Response: The monoclonal antibodies used in this work were previously published (Monoclonal Antibodies against Cell Wall Chitooligomers as Accessory Tools for the Control of Cryptococcosis) and we will reference them directly in the materials and methods. Murine antibodies were used, as humanized antibodies originating from murine antibodies are still in the development phase. 

Section 3. Results.  

Why did the authors measure the cytotoxicity of nanoparticles without amphotericin B against fungi only? Why wasn't this done for mammalian cells? 

Authors Response: The functionalized NP is not showing a toxic effect when tested against mammalian cells, which means that empty NP will not show a toxic effect. References will be added to articles that demonstrate that empty PCL and PLA-NP do not have a cytotoxic effect on mammalian cells. 

Therefore, I would like to request 30 days to correct the manuscript and I would like to thank you again for your contribution.

Regards,

Alexandre Figueiredo

Reviewer 3 Report

1.       Please enlist the same drug containing marketed formulation in the introduction and highlights their shortcomings

2.       Reviewer would like to see recent literature on similar studies in the introduction. Please enrich the same

3.       Please include the schematic representation for development procedure for NP preparation for better understanding

4.       Please include SEM\TEM image for NPs

5.       It is worth to include size and zeta potential distribution graphs in the revised manuscript

6.       What happen with the drug release profile. Please include

7.       Also talk on the stability of NP as well as mabs during the preparation or after the preparation.

8.       Author need to add limitations of this studies

9.       Future work including pk study should be discussed in future perspective of this study 

1.       Line 80: include separate material section

2.       Line 84: what was the concentrations of polymers used for NP preparation ?

3.       What was the volume of acetone

4.       Please mention stirring RPM in method of NP preparation

5.       Line 103: 500 uL or mL. verify

6.       Please check manuscript for annotations

7.       Line 106: rewrite the sentence

8.       Line 117: correct co2

9.       Line 133: correct the reference style. Seems author forgot to check the reference style. Make consistence.

10.   Line 139: Correct reference style.

11.   Line 144: correct to mL

12.   Line 144: Change the reference location

13.   Line 148: Correct the reference style

14.   Line 150: either keep h or hours

15.   Line 145 either keep wavelength of I, make it consistence in throughout the manuscript

16.   Line 164: Correct the sentence.

17.   Line 171: What is NPPS

18.   Figure 1: Please provide clear foot note for figure, its very confusing

19.   Line 177: correct to mL . Please do it in whole manuscript

20.   Manuscript contains so many typo errors. Please correct it 

Author Response

Dear Reviewer,

The authors would like to express their gratitude to the reviewers for their time and effort in offering thoughtful and insightful feedback on the manuscript. The comments and suggestions provided have notably enhanced the quality of the manuscript.

Comments

  1. Please enlist the same drug containing marketed formulation in the introduction and highlights their shortcomings
  2. Reviewer would like to see recent literature on similar studies in the introduction. Please enrich the same
  3. Please include the schematic representation for development procedure for NP preparation for better understanding

Authors Response

A schematic representation of the manufacturing process was added to the “Polymeric Nanoparticles: Preparation and Physicochemical Characterization” section.

Figure 1. Schematic representation of the nanoprecipitation method used to prepare nanoparticles.

  1. Please include SEM\TEM image for NPs

Authors Response:  As requested, Figure 4 was added to provide TEM images fot the NPs.

The following statement was added to section “Polymeric Nanoparticles: Preparation and Physicochemical Characterization”.

The morphology of the obtained nanoparticles was analyzed on the Electron Microscopy Platform of the Gonçalo Moniz Institute (IGM-FIOCRUZ/BA). 10 µL of each sample were dripped onto formvar degrees for 1 minute and subsequently incubated with uranyl acetate 2%. The excess of the contrasting solution was removed from the grid with filter paper and the samples were kept at room temperature until specific analysis.

  1. It is worth to include size and zeta potential distribution graphs in the revised manuscript

Authors Response: As requested, Figure 3 was added to provide an overview of the particle size and zeta potential distribution.

  1. What happened with the drug release profile. Please include

Authors Response: Thanks for the note. In fact, release tests are very important in the development and evaluation of drug carriers. In this specific case, the idea was to carry out initial tests that could serve as proof of concept, in order to evaluate the effect of nanoparticles in an in vitro model. There is a complete physicochemical and biopharmaceutical characterization, which includes release tests, which will form another publication, since the purpose of the current manuscript was fully achieved even without the release being properly quantified. It is assumed that, once the action has been detected, it is because the active ingredient has been released. Furthermore, as it is a differentiated delivery system, a specific publication will be made addressing the development of the analytical method used in the release test, since several parameters need to be considered in order to have a robust and discriminative method. This data would be outside the scope of the current submission.

  1. Also talk on the stability of NP as well as mabs during the preparation or after the preparation.

Authors Response: The authors acknowledge the reviewer's insightful suggestion regarding the crucial role of AmB stability in formulation efficacy. Given the well-documented instability of AmB in aqueous media, all AmB-loaded nanoparticles (AmB-NP) were freshly prepared prior to any evaluation to ensure the reliability of results. Thus, the following statement was added to section “Polymeric Nanoparticles: Preparation and Physicochemical Characterization”.

To ensure the reliability of experimental results, all amphotericin B-loaded nanoparticles (AmB-NP) were freshly prepared prior to any subsequent evaluation. This approach addressed the well-documented instability of AmB in aqueous media, mitigating potential degradation that could compromise the performance of the nanoparticles.

It is important to emphasize that mAbs was not added during the nanoparticle manufacturing process. To make it clear, section 3.3.2 Antifungal activity with AmB-loaded NP in synergy with MAbs was rewritten as follows:

To access the synergistic effect of AmB-NP with mAbs, freshly prepared nanoparticles containing non-inhibitory concentrations of PLA-AmB and PCL-AmB formulations (i.e., 0.06 μg/mL to C. neoformans and 0.03 μg/mL to C. albicans) were supplemented with mAbs. As previously mentioned, no discernible inhibition of fungal growth was observed when AmB-NP were tested independently, i.e., without association with mAbs (Figure 6).

  1. Author needs to add limitations of this studies

Authors Response:This topic was addressed together with suggestion N° 9 in a section named “Future Perspectives.

  1. Future work including pk study should be discussed in future perspective of this study.

 The authors agree that by including a discussion about the current limitations of this study and its future perspective, **. Therefore, a new section was added to the manuscript named “Future Perspectives.

  1. Future perspectives

This study aimed to establish a proof-of-concept by demonstrating the synergistic effect of AmB-loaded NP in combination with mAbs against fungal cells. The observed data support the potential of these formulations for antifungal therapy. Nevertheless, further development studies are required to translate this concept into a clinical applicable formulation. Freeze-drying, for example, appears to be a promising approach to mitigate AmB aqueous instability, potentially improving long-term storage and transportability. Additionally, in vivo studies, e.g., pharmacokinetic – pharmacodynamic (PK/PD) modeling, can provide valuable insights into the relationship between drug exposure and antifungal activity. Implementing these models will facilitate the optimization of dosing strategies and treatment schedules for optimal therapeutic efficacy.

Once again, I thank you for all the notes that will enrich the manuscript.

Therefore, I would like to request 30 days to correct the manuscript and I would like to thank you again for your contribution.

Regards,

Alexandre

Round 2

Reviewer 2 Report

The authors fully answered the reviewer's questions and comments. The reviewer believes that the efforts made by the authors to revise the manuscript have significantly improved its quality and the manuscript can be accepted for publication.

No comments